# Localizing Sustainable Development Goal 6: An Assessment of Equitable Access to Sanitation in a Brazilian Metropolitan Region

**Rodrigo Coelho de Carvalho [1],\*, Maria Inês Pedrosa Nahas [2]**  **and Léo Heller [2]**

[1]  Center for Development and Regional Planning—Cedeplar, Faculty of Economics—Face, Federal University of Minas Gerais—UFMG, Belo Horizonte 31270-901, Brazil

[2]  René Rachou Institute—IRR, Oswaldo Cruz Foundation—Fiocruz, Belo Horizonte 30190-009, Brazil; maria.nahas@fiocruz.br (M.I.P.N.); leo.heller@fiocruz.br (L.H.)

\*  Correspondence: mailrcarvalho@cedeplar.ufmg.br

**Abstract:** In order for the goals and targets of the 2030 Agenda to be achieved, it is essential to "localize" the Sustainable Development Goals (SDGs), since it is only at the local level that it is possible to move towards their effective implementation. This article seeks to contribute to the development of evaluation and monitoring strategies for target 6.2 at the local level, adapting the official SDG indicator 6.2.1a and the international criteria established by the Joint Monitoring Programme for Water Supply, Sanitation and Hygiene (coordinated by the World Health Organization (WHO) and the United Nations Children's Fund (UNICEF)) at the municipal scale. Using the Belo Horizonte Metropolitan Region (RMBH) as a case study, a series of methodological procedures is proposed to assess and monitor equitable access to sanitation services. Inequalities in access to services between different population subgroups and between the municipalities that make up the RMBH are explored in different ways, including the evaluation of intersecting forms of inequality, the mapping of a synthetic index of inequality based on multiple criteria and the projection of the time needed to achieve universal access to services according to international criteria. The procedures applied demonstrated the existence of significant inequalities among the municipalities and population subgroups of the RMBH, which are not evident in the analysis of the aggregated data by municipality.

**Keywords:** SDG 6; target 6.2; localizing; sanitation; inequality; RMBH

## 1. Introduction

The diagnosis of the sanitation situation in Brazil presented in the National Plan of Basic Sanitation (PLANSAB) showed great inequalities in access to services [1], reflecting the intense socio-spatial segregation observed in several major Brazilian cities [2–5], as is the case in the Belo Horizonte Metropolitan Region (RMBH) [6], the third-largest metropolitan region in the country. Since its founding in 1897, the history of the city of Belo Horizonte has been marked by a systematic gap between population growth and the capacity of water supply and sanitation systems, especially in its peripheral neighborhoods [7]. In addition, the accelerated and disordered process of metropolization that has occurred in the country since the middle of the last century has led to the formation of highly deficient public service systems, especially in metropolitan outskirts. In the case of the RMBH, established in 1973, this problem is reflected in the absence of sanitation services and inequalities in access between population subgroups and among the 34 municipalities that currently comprise the region. Although the RMBH is located in the most economically developed region of the country (Brazilian Southeast) and its population as a whole has a higher coverage of sanitation services than

the national average [6], the level of access varies widely between municipalities and population subgroups. According to the 2010 Demographic Census, almost seven thousand households in the RMBH did not have bathrooms, a pervasive problem in many of the informal settlements existing in the region, where housing precariousness is particularly evident [8].

As sanitation is a social and environmental determining factor of public health [9], it was incorporated in different ways in the 17 Sustainable Development Goals (SDGs) and 169 targets of the 2030 Agenda, signed at the UN Sustainable Development Summit in 2015. SDG 6 is specifically concerned with drinking water, sanitation, and hygiene (WASH), referring to the need to "ensure availability and sustainable management of water and sanitation for all". It comprises eight targets, the second one (6.2) being of particular interest for the purposes of this work: "By 2030, achieve access to adequate and equitable sanitation and hygiene for all and end open defecation, paying special attention to the needs of women and girls and those in vulnerable situations". Meeting this target is extremely relevant since sanitation and hygiene for all is a major contributor for achieving other targets of the Agenda, such as those related to health, poverty, education, and inclusive cities.

Although the SDGs are global, their achievement will depend on the ability to make them a reality in cities and regions and, for that, "localizing" them is essential. The "localization" process of the SDGs consists of defining implementation and monitoring strategies at the local level to achieve the goals and targets set forth in the 2030 Agenda [10,11]. This requires a multilevel governance coordination "to ensure cross-scale integration and the design of mutually supportive and cohesive national and territorial policies" [12] (p. 1). Regional governments, for example, play a key role in bridging different levels of public administration and sectoral agencies, building on trends and guidelines from national and global levels and assisting local governments and municipalities [12]. Despite the undeniable importance of national and regional governments for the achievement of the SDGs, this will only be possible with the support of local governments, since actions are more likely to produce effective and measurable results at the local level [12–14].

There is a vast literature about localizing SDGs in general and a plethora of case studies regarding specific cities and local actions. Nevertheless, despite their interdependent natures, certain SDGs are receiving much more attention than others. A great amount of research was dedicated, for example, to the localization of SDG 11 ("Sustainable cities and communities"), including in the RMBH [15] (the website of the Institute for Housing and Urban Development Studies of Erasmus University Rotterdam, for example, provides an extensive list of publications on localizing SDG 11 [16]). On the other hand, not many attempts have been made to localize SDG 6, with a few exceptions, such as the work of Patole, where 6 was used as an example to "highlight methodologies for localization, disaggregation, and development of inter-linkages between SDGs to develop new key performance indicators" [17] (p. 6). This neglect is clear in a survey of 15 cases showing successful implementation of local actions in cities across Europe, in which SDG 6 appears only once—and in conjunction with six other SDGs [14].

The official United Nations baseline for the global assessment and monitoring of target 6.2 was the report launched in 2017 by the Joint Monitoring Programme for Water Supply, Sanitation and Hygiene (JMP) [18]. The JMP is jointly coordinated by the World Health Organization (WHO) and the United Nations Children's Fund (UNICEF) and it has reported country, regional, and global estimates of progress on WASH since 1990, being the leading source of comparable estimates of progress at these levels [19]. However, the definition of local or regional plans requires the adaptation of the JMP methodology and the use of indicators adjusted to the possibilities, needs, and context of each territory.

Against this background, this paper seeks to contribute to the process of "localizing" SDG 6 by proposing and applying a methodology for assessing and monitoring the access to equitable sanitation services, using the RMBH as a case study. Access to services by different population subgroups and by the municipalities that compose the RMBH is explored. The use of municipalities as units of analysis is justified because, besides being the lowest level of public administration, they often have a single provider of WASH services. Thus, by comparing municipalities, the analysis includes the supply-side

of the services and, by comparing subgroups within them, the analysis encompasses the inequalities on the demand-side. In addition to assessing socio-spatial inequalities, an estimate of the current situation in this region as a whole is presented based on the most recent data available and a projection of universal access to sanitation services according to international criteria, based on the trends observed in recent decades.

## 2. Materials and Methods

### 2.1. Materials

In Brazil, the estimation of the levels of access to services according to the conceptual framework proposed by the JMP [18] is only possible with census data, which also allows international comparison. JMP uses a sanitation "service ladder" made up of five "steps" as a reference to assess, monitor, and compare progress between countries and regions. Estimates of levels of access to services combine the classification of "facility types" (Table 1) with some attributes related to services. In addition to "safely managed" services, the ladder comprises the "basic", "limited", "not improved", and "open defecation" levels [18], described in Table 2.

**Table 1.** Joint Monitoring Programme for Water Supply, Sanitation and Hygiene (JMP) classification of improved and unimproved facility types.

| Improved Facilities | Networked sanitation: | |
|---|---|---|
| | • Flush and pour flush toilets connected to sewers | |
| | On-site sanitation: | |
| | • Flush and pour flush toilets or latrines connected to septic tanks or pits<br>• Ventilated improved pit latrines<br>• Pit latrines with slabs<br>• Composting toilets, including twin pit latrines and container-based systems | |
| Unimproved Facilities | On-site sanitation: | |
| | • Pit latrines without slabs<br>• Hanging latrines<br>• Bucket latrines | |
| No Facilities | Open defecation | |

Source: World Health Organization (WHO)/United Nations Children's Fund (UNICEF)/JMP, 2017 [18].

**Table 2.** JMP ladder for sanitation services.

| Service Level | Definition |
|---|---|
| Safely Managed | Use of improved facilities that are not shared with other households and where excreta are safely disposed of in situ or transported and treated offsite |
| Basic | Use of improved facilities that are not shared with other households |
| Limited | Use of improved facilities shared between two or more households |
| Unimproved | Use of pit latrines without a slab or platform, hanging latrines, or bucket latrines |
| Open Defecation | Disposal of human feces in fields, forests, bushes, open bodies of water, beaches, or other open spaces, or with solid waste |

Source: WHO/UNICEF/JMP, 2017 [18].

The indicator "proportion of the population using safely managed sanitation services" (indicator 6.2.1a) was one of the global indicators formulated by the Inter-Agency and Expert Group on SDG Indicators (IAEG-SDGs) to encompass in a synthetic way the different components of target 6.2, i.e., "access to adequate and equitable sanitation for all, paying special attention to the needs of those in

vulnerable situations". However, there are no data available in Brazil to measure access to services at this level of detail. The biggest challenge is the deficiency or lack of information related to the disposal and treatment of effluents, excreta, and sump sludge, especially at the municipal and intra-municipal scale. As a result, for the purpose of this work, the concept of "at least basic" services, also used by the JMP, was adopted. This classification is used when there is no sufficient data to distinguish the "basic" level from the "safely managed" level. It corresponds to the second-highest level of the ladder ("basic") and includes the population that meets the criteria of the first level ("safely managed"). In Brazil, the only database that allows estimating the proportion of the population of municipalities with access to this level of sanitation services is the Demographic Census, which has the additional advantage of allowing data disaggregation by population subgroups. As in the national estimates made by the JMP, only residents of permanent private households were considered [18].

Since the last Brazilian census was carried out ten years ago, the assessment of the current access to sanitation in the RMBH was complemented by two household sample surveys from the Brazilian Institute of Geography and Statistics (IBGE): National Household Sample Survey (PNAD) and Continuous PNAD ("PNAD Contínua"). Despite having a similar name, these are distinct surveys, with the latter replacing the former definitively in 2016 [20]. Among several methodological changes and improvements, the Continuous PNAD (implementation of which started in 2012) has a much broader scope of data collection: while the PNAD included 1100 municipalities, the Continuous PNAD includes 3500 of the current 5570 municipalities in Brazil; while the PNAD included only nine metropolitan regions (the highest level of spatial disaggregation of this survey), the Continuous PNAD includes all twenty metropolitan regions containing state capitals and the state capitals' municipalities (thus, adding an additional layer of spatial disaggregation). In addition, the Continuous PNAD follows a system of household rotation, where each selected household is visited five times, once each quarter, for five consecutive quarters. This sample design represents an improvement on the periodicity of the survey, although the frequency of the researched themes or topics varies (there are permanent research topics, investigated on a regular basis and supplementary ones, included in the survey with variable frequency).

*2.2. Methodological Procedures*

To assess and monitor access levels and socio-spatial inequalities regarding sanitation services in the RMBH, a series of methodological procedures were adopted, as described below.

2.2.1. Estimation of the Current Situation of Access to Sanitation Facilities in the RMBH as a Whole

- The PNAD and Continuous PNAD were used to trace the evolution of access levels to sanitation facilities in the RMBH until the most recent period available. It was possible to estimate for each year of the period 2010–2018 the access levels to sewerage or septic tanks connected to sewer systems (both classified as "improved facilities" by the JMP) and to private toilets in the households of the RMBH as a whole (the criterion that distinguishes the "limited" and "basic" levels). Therefore, the following procedures were followed. The "proportion of the population with access to sewerage or septic tank connected to sewer system" was estimated for 2010 from the Demographic Census of that same year. This variable was computed by adding the population with access to sewerage to 37% of the population using septic tanks. This correction in the proportion of the population with septic tanks is equivalent to the proportion of users of septic tanks connected to sewer systems, according to the 2011 edition of the National Household Sample Survey (PNAD). This procedure was necessary, as the 2010 Census makes no distinction between septic tanks connected and those not connected to sewer systems. From 2011 to 2015, data from PNAD were used and, from 2016 to 2018, data from the Continuous PNAD were used, referring to the variable "sewerage or septic tank connected to sewer system".
- The variable "proportion of the population with access to a private toilet" was composed of different data sets, including the variable "private toilet" of the 2010 Demographic Census;

"toilet or private bathroom" of the PNADs, from 2011 to 2015; "bathroom, toilet or hole for dejections of exclusive use" of the 2016 edition of the Continuous PNAD; and "private toilet" of the 2017 and 2018 editions of the Continuous PNAD.

In order to compensate these sampling and methodological variations across the period and better represent the general trend of access to sanitation facilities in the RMBH, a simple moving average was applied to the data.

### 2.2.2. Estimation of Access to "At Least Basic" Sanitation Services in the Municipalities of the RMBH and by Different Population Subgroups

The comparison of access levels between the RMBH municipalities and between different population subgroups was performed by disaggregating four variables from the 2010 Demographic Census microdata (sample questionnaire): urban–rural status of the household; educational attainment for those individuals aged 25 or older (assuming the time required to complete higher education); classes of household income per capita (in minimum wages); and color or race (white and non-white). Although the standard unit of analysis of sanitation variables is the household, the values were applied to the residents, individually. This is important because, due to the tendency of larger households to have more unfavorable socioeconomic situations, analysis at the household level may show an overestimated coverage of sanitation facilities and services [21]. Although it is a crucial ground of discrimination, gender was not included among the criteria, as it is primarily an intrahousehold dimension of inequality.

In the classification of "at least basic" services, in addition to the use of improved facilities, the use of a private toilet must also be taken into account, because this is the criterion that distinguishes "limited" and "basic" levels. Among census categories, only "sewerage" and "septic tanks" were classified as "improved facilities", as shown in Table 3. In the case of the "traditional latrines", as this does not distinguish pit latrines with and without slabs (necessary for the distinction of the type of facility), the same methodological procedure used in the national JMP estimates was applied: 50% of the facilities in this category were considered "improved" and 50% were considered "unimproved". The categories "ditch", "river, lake or sea", and "other" were classified as "unimproved" or "no facility".

**Table 3.** Adaptation of the JMP facility types classification.

| Type of Facility (JMP Classification) | Census Variable (in Portuguese) | Classification |
|---|---|---|
| Flush/toilet to piped sewer system | "Rede geral de esgoto ou pluvial" | Improved facility |
| Flush/toilet to septic tank | "Fossa séptica" | Improved facility |
| Traditional latrine | "Fossas rudimentares" | 50% improved and 50% unimproved |
| Pit latrine without slab/open pit and others | "Vala, rio, lago ou mar e outro" | Unimproved/no facilities |

### 2.2.3. Projection of the Year of the Universalization of "At Least Basic" Sanitation Services in the RMBH Municipalities

In order to assess the pace of progress (or setbacks) in the levels of access to "at least basic" sanitation services in the RMBH municipalities, a projection of the time required for coverage universalization of this level of services in each municipality was made, considering the maintenance of the same growth rate observed in the 2000–2010 inter-census period. To carry out this prospective exercise, a linear growth rate was considered up to the point of universalization. Although the projection of reaching target 6.2 would be ideal, it would require sufficient data to estimate the "proportion of population using safely managed sanitation services", as established in indicator 6.2.1a. As this is not the case, the year of access universalization to "at least basic" services in the municipalities of the RMBH was used as a proxy. It should be taken into account that, as the "safely managed" criteria are stricter

than the "at least basic" one, the projected time to reach target 6.2 should be even longer than the one presented in the results.

### 2.2.4. Inequality Index Formulation

A synthetic inequality index was created to capture multiple dimensions of inequality in access to sanitation services. It was based on the disaggregation of three census variables: household income per capita, educational attainment, and color or race. The urban–rural status of the household was not considered because the inclusion of this criterion could be misguiding, since the rural population is residual or even non-existent in several municipalities of the RMBH. The calculation of the inequality index was based on the ratio between the proportion of access to "at least basic" sanitation services of the "least vulnerable" and the "most vulnerable" group (Table 4). In the case of color or race, the ratio between access to services by whites and blacks or browns was considered. In the case of educational attainment, it was based on the ratio between access to services by the population with university completed and the population with no formal education or less than primary education completed (considering only people with at least 25 years of age). In the case of income, it was based on the ratio between access to services by the population residing in households with per capita income equal to or greater than one minimum wage and those residing in households with per capita income below one-quarter of the minimum wage. A per capita income below a quarter of the minimum wage was considered as a proxy for the poverty line. This threshold, also used by the Institute of Applied Economic Research (IPEA), other Brazilian official government agencies, and researchers was defined based on the official poverty line established by the federal government in 2011, in the Plan to Overcome Extreme Poverty ("Plano de Superação da Extrema Pobreza")—known as "Brazil Without Misery" ("Brasil Sem Miséria")—and it is related to the international line defined by the World Bank [22].

**Table 4.** Criteria for defining the vulnerable and non-vulnerable group.

| Variables | "Vulnerable" Group | "Non-Vulnerable" Group |
|---|---|---|
| Color or race | Black or brown | Whites |
| Educational attainment | No formal education or less than primary education completed | University completed |
| Household income (per capita) | Less than 1/4 minimum wage | Equal to or greater than one minimum wage |

As three criteria were considered, the three ratios were summed and the result was divided by that number, as shown in the formula below:

$$ID = (W/B + U/N + H/L)/3, \tag{1}$$

where ID corresponds to the Inequality Index and the remaining terms to the proportion of population groups with access to "at least basic" sanitation services: W (white); B (black and brown); U (university completed); N (no formal education or less than primary education completed); H (household income per capita equal to or greater than one minimum wage); L (household income per capita less than a quarter of a minimum wage). Under ideal conditions, without discrimination and inequality, the result would be equal to 1.0. In cases where the group presumably less susceptible to vulnerable situations has higher levels of access, the result would be greater than 1.0 and, the higher the value, the greater the level of inequality.

Mapping the Inequality Index allowed for the simultaneous exploration of inequality levels between the RMBH municipalities and between the population subgroups residing in them. Five classes of municipalities were created using the "natural breaks" method or "Jenks optimization method",

which seeks to minimize the variance within each class while seeking to maximize the variance between classes. The maps and classes were created using ArcMap software by ESRI, 10.2.2 version.

### 2.2.5. Intersectional Analysis of Inequalities in the Access to Sanitation Services in the RMBH Municipalities

As vulnerable people often present more than one ground of discrimination at the same time [23], their chances of deprivation of basic public services can be exacerbated, as in the case of sanitation services. For this reason, the effects of intersecting forms of inequality [23] were examined by comparing the level of inequality between two antagonistic population profiles with regard to susceptibility to vulnerable situations: white people with household income per capita equal to or greater than a minimum wage and university completed and black or brown people with household income per capita below a quarter of a minimum wage and no formal education or less than primary education completed. As sample sizes of these two groups tend to be quite different in the RMBH municipalities, in order to avoid distortions in the comparison, municipalities where one of the two groups represented less than 1% of the total population (six cases in the first group) and municipalities with no sewerage (two cases) were disregarded.

## 3. Results

### 3.1. Current Situation of Access to Sanitation Facilities in the RMBH as a Whole

Figure 1 shows the estimates of the proportion of the RMBH population with access to sewerage or septic tank connected to the sewer system and with access to a private toilet in the household from 2010 to 2018. Although access to private toilets has been practically universal since the beginning of the period considered, the proportion of residents of households connected to sewerage is much lower. The data suggest a general trend of increase in the proportion of people with access to sewerage since the 2010 Demographic Census, reaching a coverage of 92.2% of the population according to the 2018 edition of the Continuous PNAD (disregarding the moving average).

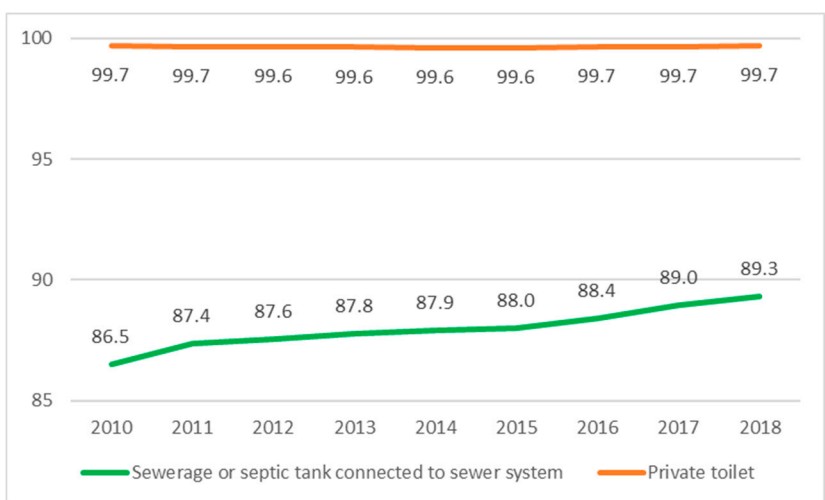

**Figure 1.** Proportion of residents of permanent private households in the RMBH with access to sewerage or septic tank connected to sewer system and private toilets in the household (own elaboration based on a simple moving average of data from the 2010 Demographic Census, National Household Sample Survey (PNAD) 2011–2015, and Continuous PNAD 2016–2018).

*3.2. Access to "At Least Basic" Sanitation Services in the Municipalities of the RMBH and by Different Population Subgroups*

Figure 2 shows the proportion of the population of the RMBH municipalities with access to "at least basic" sanitation services in 2010. The municipality of Belo Horizonte, capital of the state of Minas Gerais, had the highest coverage and, although the graph already shows the disparity between the metropolitan core and some peripheral municipalities, these aggregate values mask significant intra-municipal inequalities, as shown in the following figures.

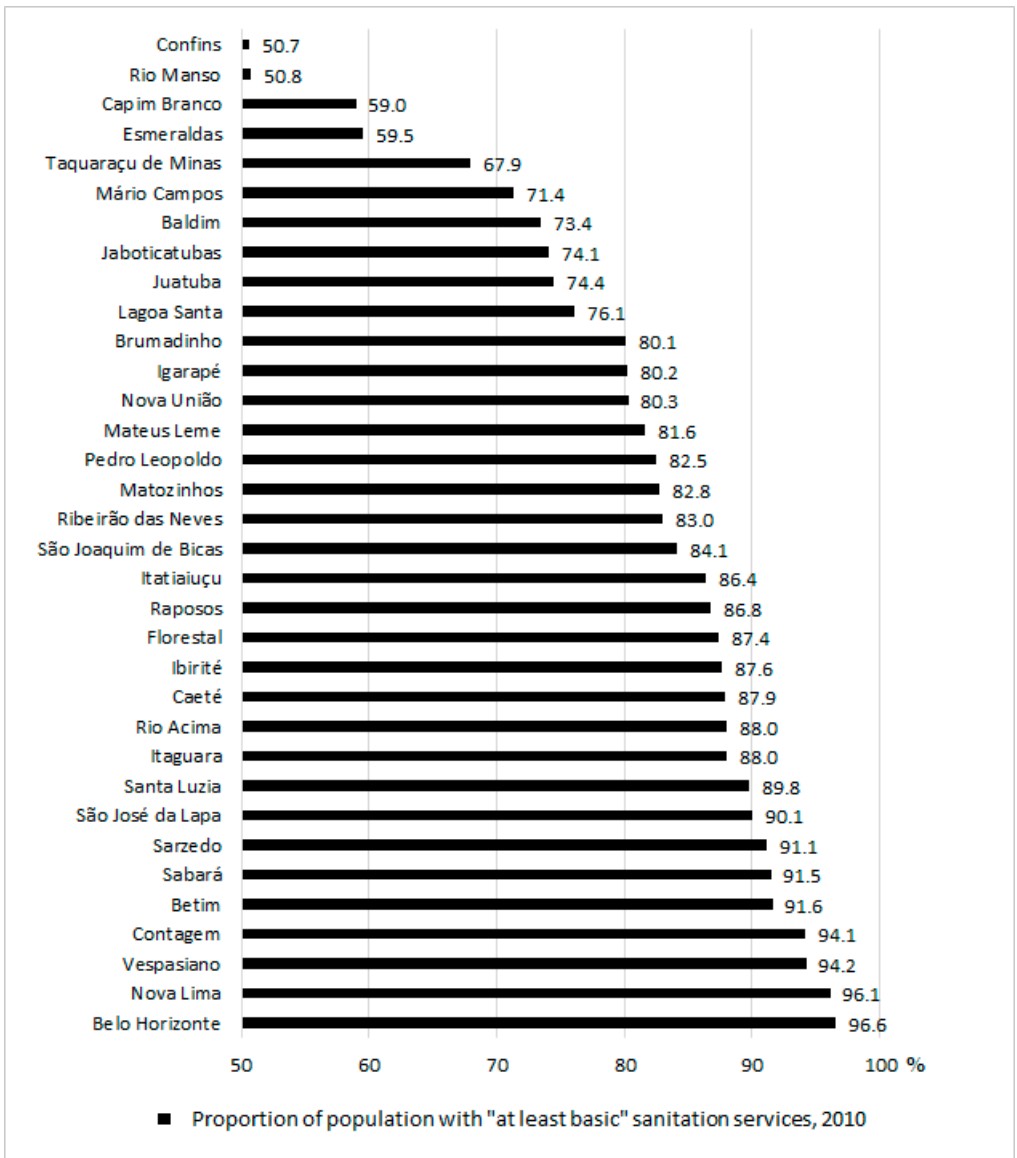

**Figure 2.** Proportion of the population with access to "at least basic" sanitation services in 2010 (own elaboration based on data from the 2010 Demographic Census).

Figure 3 shows the proportion of the rural and urban populations of the RMBH municipalities with access to "at least basic" sanitation services in 2010. There is extreme inequality between urban and rural areas: in the RMBH as a whole, the difference was more than 26 percentage points (p.p.) and, in 12 of the 34 municipalities, the difference exceeded 30 p.p., reaching 60 p.p. in one of the municipalities.

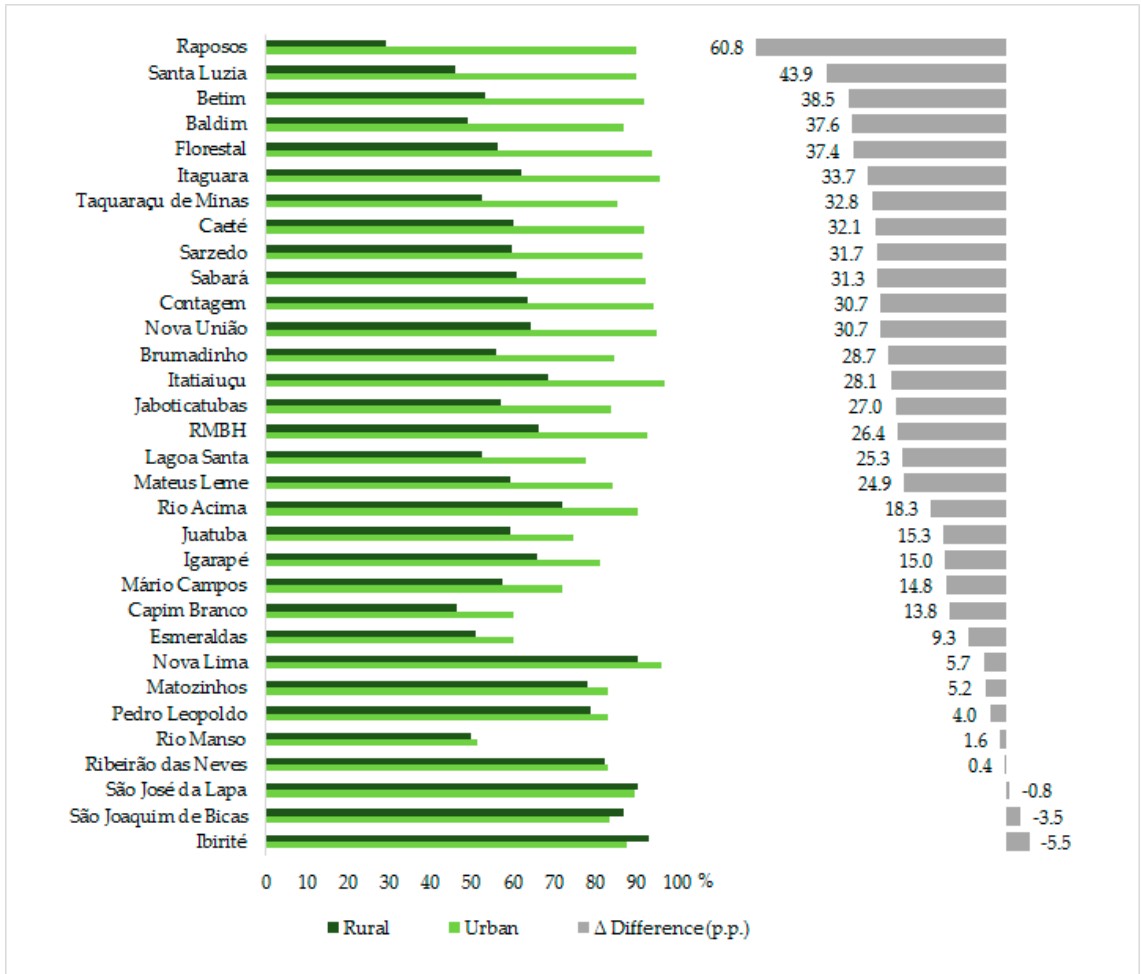

**Figure 3.** Proportion of the rural and urban population with access to "at least basic" sanitation services in 2010 (authors' own elaboration based on data from the 2010 Demographic Census). p.p.—percentage points.

Figure 4 shows the proportion of the population with access to "at least basic" sanitation services according to color or race in 2010, divided into two categories: white and non-white. All municipalities showed differences in access levels due to color or race in favor of white people, except in three cases, which presented an atypical pattern, favorable to non-whites. In the RMBH as a whole, the inequality between whites and non-whites is almost four percentage points. In one municipality, the difference between the two groups exceeded nine percentage points.

Figure 5 shows the proportion of the population with access to "at least basic" sanitation services according to their educational attainment in 2010. The "equiplot" graph shows, in addition to the coverage levels of "at least basic" services, the inequalities between the four educational levels within each municipality. The municipalities were ordered accordingly to the level of inequality between the two extreme categories, that is, by the difference between people with university completed and people with no formal education or less than primary education completed with access to services. Predictably, levels of access to "at least basic" services tend to increase according to the level of education, but in some municipalities this increase is particularly pronounced. In four municipalities, the level of inequality between the extreme categories exceeded 20 percentage points (reaching 36 p.p. in one of the municipalities). In the most populous municipalities of the RMBH, the progression of access levels depending on the level of education is particularly evident.

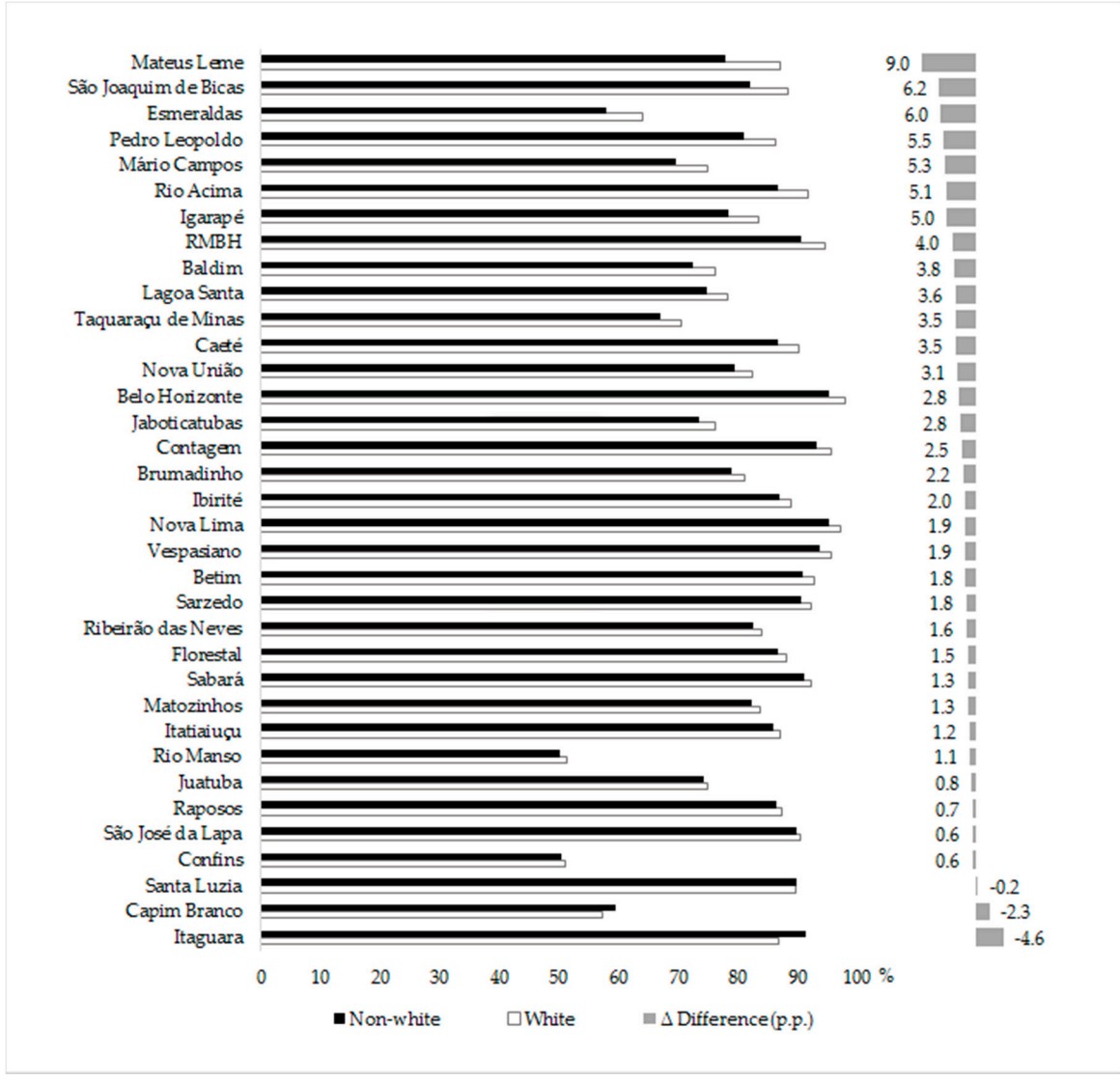

**Figure 4.** Proportion of the population with access to "at least basic" sanitation services according to color or race in 2010 (authors' own elaboration based on data from the 2010 Demographic Census).

The equiplot in Figure 6 shows the proportion of the population with access to "at least basic" sanitation services according to household income per capita in 2010. Once again, municipalities were ordered by the level of inequality between the extreme categories, i.e., the difference in access levels between residents of households with income per capita equal to or higher than two minimum wages and households with income per capita below one-quarter of the minimum wage. While in the RMBH as a whole, the difference in access between extreme categories is 11.2 p.p., in five municipalities this difference exceeds 20 p.p., reaching more than 30 p.p. in two of them. At the other extreme, four municipalities showed insignificant differences between these categories, less than 2 p.p.

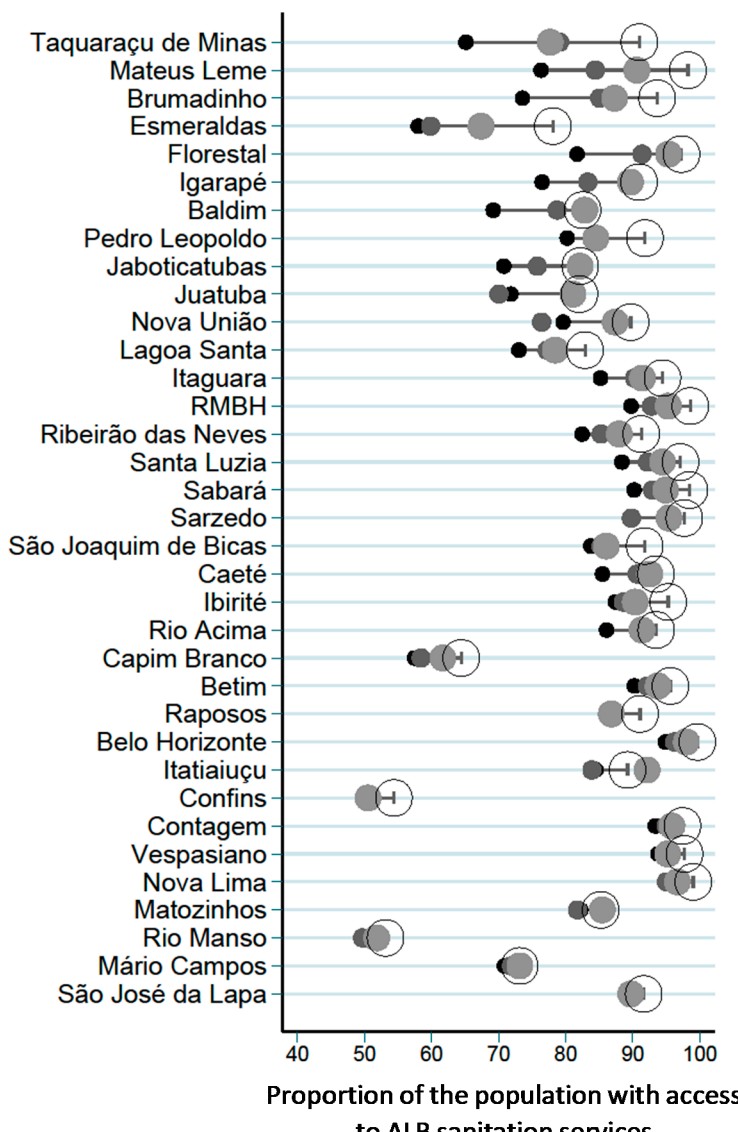

**Figure 5.** Proportion of the population with access to "at least basic" (ALB) sanitation services according to educational attainment (considering only people 25 years old or older) in 2010 (authors' own elaboration based on data from the 2010 Demographic Census).

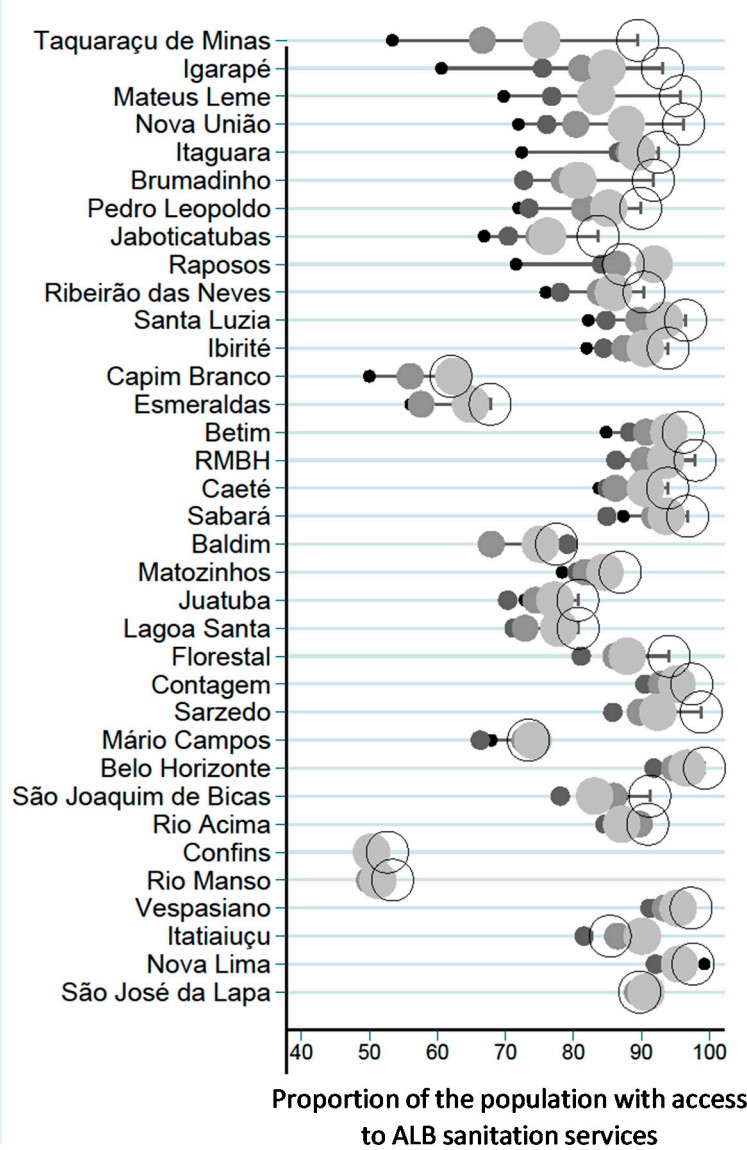

Household income (*per capita*):

    ◯   Equal to or greater than 2 minimum wages (m.w.)

    ⬤   Equal to or greater than 1 m.w. and less than 2 m.w.

    ⬤   Equal to or greater than ½ m.w. and less than 1 m.w.

    ⬤   Equal to or greater than ¼ m.w. and less than ½ m.w.

    ●   Less than ¼ m.w.

**Figure 6.** Proportion of the population with access to "at least basic" (ALB) sanitation services according to household income per capita in 2010 (authors' own elaboration based on data from the 2010 Demographic Census).

*3.3. Universalization of "At Least Basic" Services in the RMBH Municipalities*

Figure 7 shows the projection for the year of the universalization of "at least basic" sanitation services in the RMBH municipalities, assuming that the pace of expansion observed in the inter-census

period between 2000 and 2010 is maintained. Considering a linear growth rate, six municipalities in the RMBH would have already achieved universalization, including the two most populous municipalities in the RMBH after the metropolitan core. Considering the same assumptions, nine other municipalities—including Belo Horizonte—should reach universal access within this decade. Although only 15 of the 34 municipalities in the RMBH are expected to reach this point by 2030, it is worth mentioning that the most populous municipalities in the region are included in this set. Their population was equivalent to almost 90% (89.1%) of the total population of the RMBH in the year 2010, according to the Demographic Census of that same year. The remaining municipalities would reach universalization only after 2030, due to their low levels of access and/or the very slow pace of expansion of services. Maintaining the same conditions, in four of the municipalities represented in the graph, universalization would be achieved only by the next century. However, the most serious situations were observed in three municipalities (not shown in the graph) that showed a retraction in access levels between 2000 and 2010, which may represent a setback in the fulfillment of the human right to sanitation.

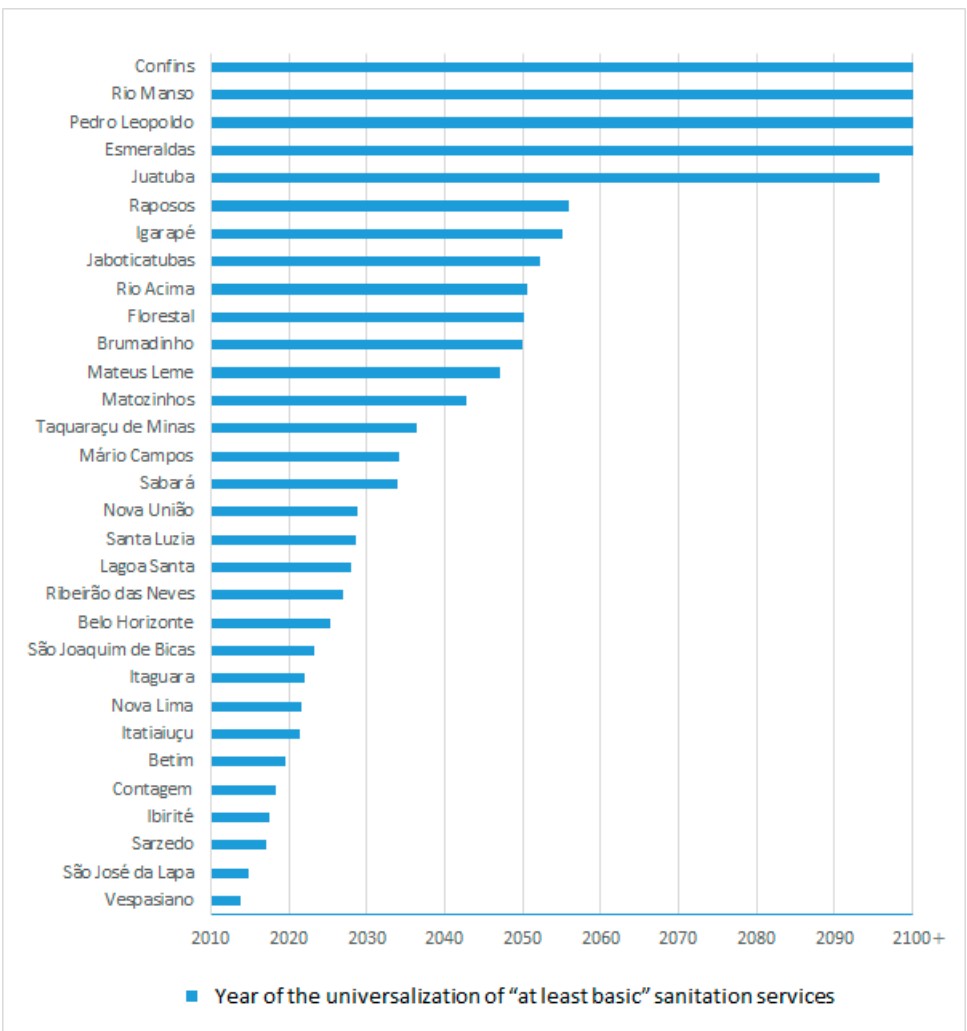

**Figure 7.** Projection of the universalization of access to "at least basic" sanitation services (authors' own elaboration based on the 2000 and 2010 Demographic Censuses microdata).

### 3.4. Inequality Index

The map in Figure 8 represents the Inequality Index for access to "at least basic" sanitation services in the RMBH municipalities. With the exception of a single municipality that had an index equal to one (that is, absence of inequality), all the others had some degree of inequality. In the most extreme case,

the Inequality Index reached 1.3, which means that, on average, the most favored groups (in terms of education, income, and color or race) had levels of access to "at least basic" services 30% higher than the least favored groups. Apart from the fact that the municipalities classified in the worst category are relatively distant from Belo Horizonte and those classified in the best category are located immediately in the North and South of that municipality, the map does not show any clear spatial pattern on the municipal scale.

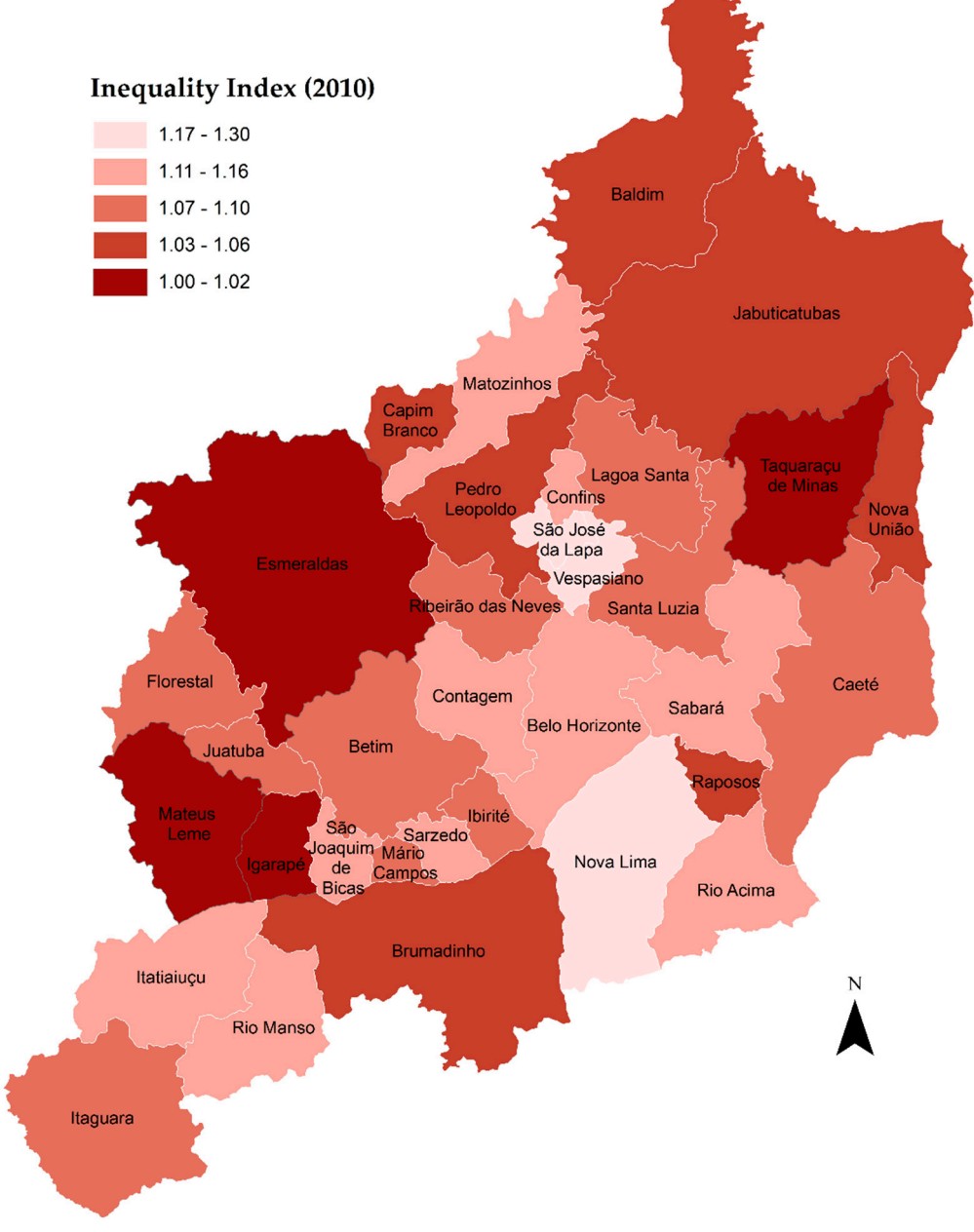

**Figure 8.** Map of the Inequality Index of access to "at least basic" sanitation services in the RMBH municipalities (authors' own elaboration based on data from the 2010 Demographic Census).

*3.5. Inequalities in the Access to Sanitation Services between Population Subgroups in the RMBH Municipalities Considering Intersecting Forms of Inequality*

Figure 9 represents access to "at least basic" sanitation services between two antagonistic population profiles regarding the susceptibility to situations of socioenvironmental vulnerability: white people with household income per capita equal to or greater than a minimum wage and

university completed and black or brown people with household income per capita below a quarter of a minimum wage and no formal education or less than primary education completed. Due to the overlapping of multiple layers of inequality, the differences observed between the two groups were much more pronounced in comparison to the previous analyses, with the exception of the analysis by urban–rural status. In three municipalities, the difference in access to "at least basic" sanitation services was greater than 30 p.p. and, of the 26 municipalities represented in the graph, 21 showed a difference between groups higher than 7 p.p.

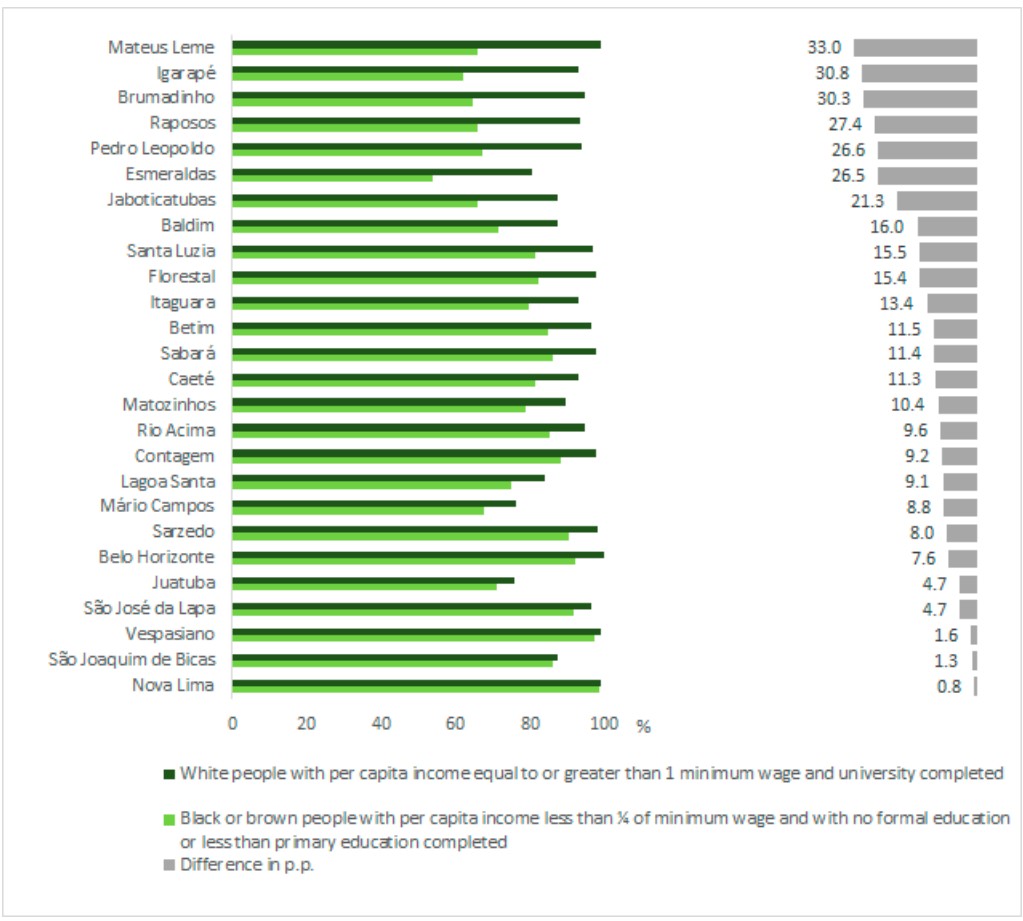

**Figure 9.** Access to "at least basic" sanitation services considering intersecting forms of inequality (authors' own elaboration based on data from the 2010 Demographic Census).

## 4. Discussion

The results presented in this paper comprise different scales of analysis, including the RMBH as a whole, its municipalities, and the subdivisions of municipal populations (disaggregated by urban–rural status, color or race, educational attainment, and household income per capita). As the level of data disaggregation increases, socio-spatial inequalities in access to sanitation services and facilities become increasingly evident, making clear that these differences are masked in the aggregate values of the RMBH and its municipalities. This shows the importance of "localizing" the SDGs for an adequate assessment and monitoring of the 2030 Agenda targets and for local governments to identify the areas and groups to be prioritized by government actions aimed at achieving these targets (in the case of this study, target 6.2).

In addition to adapting the methodology proposed by the JMP to the municipal scale, this research aimed to fill some of the gaps raised by the "Task Force on Monitoring Inequalities for the 2030 Sustainable Development Agenda" [24]. This group was created by the JMP to systematically identify

the challenges related to monitoring inequalities in access to water supply, sanitation, and hygiene services. It produced a report containing a set of recommendations referring to human rights to water and sanitation and the 2030 Agenda [24]. Among the priorities identified by the group, the need for disaggregated analyses and the use of spatial data to adequately monitor inequalities at the subnational level stands out, going beyond the simple division between urban and rural and addressing poverty and deprivation in multiple dimensions [24]. In this sense, the "Inequality Index" methodology, including its cartographic representation, contributes to the development of "new visualization tools to show different types of inequalities", also recommended by the Task Force [24].

The first results presented refer to access to sanitation facilities in the RMBH as a whole using the most recent data available from the IBGE's household sample surveys. This strategy was used to compensate for the time lag of a decade in census data and enabled the visualization of a general trend of advances, due to the increase in access to sewerage. Although apparently high, the values related to access to a private toilet in the households and the proportion of residents of households connected to sewerage refer to the total population of the RMBH, masking important differences between municipalities and population subgroups. Since the RMBH is a spatially heterogeneous area and, like other metropolitan regions in the country, marked by profound socioeconomic inequalities, disaggregated analyses are essential to assess these disparities.

Data disaggregation by urban–rural status revealed a very sharp discrepancy in the coverage of "at least basic" sanitation services. However, it is important to remember that the study area is a metropolitan region, i.e., an essentially urban area. More than half of the municipalities in the RMBH (19) have less than 10% of the population living in rural areas (or even nonexistent) and, in the region as a whole, the rural population corresponded to only 2.43% of the total population in 2010. This requires caution in the analysis. For example, in the municipality with the highest proportion of rural population with access to "at least basic" services in relation to urban areas, even exceeding urban coverage (a very atypical pattern), the rural population corresponded to only 0.23% of the total population in 2010. As in the case of this municipality, the four most populous municipalities after the metropolitan core also have a residual rural population, lower than 1% (in contrast, four municipalities had a significant rural population, greater than 40%).

In what regards inequalities between population subgroups, although disparities between whites and non-whites are comparatively less pronounced than those observed between residents of urban and rural areas, the latter can be easily explained: economies of scale in large urban centers allow an optimized allocation of resources in the provision of sanitation services. On the other hand, the difference by color or race has no justification besides being one of the most well-known grounds of discrimination, placing non-white people in a condition of vulnerability (and the interweaving of this variable with socioeconomic status should not be ruled out) [23,25].

In general, inequalities by income level were more pronounced than those related to educational attainment: in the RMBH as a whole, while the difference in access to "at least basic" sanitation services between the extreme categories of education was 8.8 percentage points, the difference between the extreme income categories was 11.2 (but the higher number of income categories must be taken into account that, because it tends to amplify the differences). Despite these differences, the level of access to "at least basic" sanitation services varies accordingly to both criteria, such that the higher the level of education and income, the higher the level of access to services. The strong connection between education and income inequality is a well-documented feature of Brazil [26]. Data from the OECD publication "Education at a Glance" shows a close correlation between level of earnings and educational attainment in the country: In 2015, Brazilian workers with a university degree had an income 149% higher in comparison with those who have only secondary education, the greatest difference among the 38 countries analyzed (34 from OECD and 4 partners) [27] (p. 99).

The projection of the year for the universalization of access to "at least basic" sanitation services showed that the pace of progress is extremely disparate between the RMBH municipalities—while some of them are moving quickly towards universal access, others are practically stagnant or even

presented a retraction in access levels. On the other hand, it is worth mentioning that the most populous municipalities in the RMBH are among those closest to achieving universal access to "at least basic" services by 2030—which is not the same as reaching target 6.2, which concerns the stricter category of "safely managed" sanitation services.

Although the Inequality Index map did not show any spatial pattern particularly evident on the metropolitan scale, it was observed that inequality levels in access to "at least basic" sanitation services tend to be less pronounced in the metropolitan core and in its immediate surroundings than in the rest of the RMBH. Of the four municipalities classified as the most unequal, none is adjacent to Belo Horizonte and, of the seven municipalities classified in the second-worst category, only one has a border with Belo Horizonte. Additionally, of the three most egalitarian municipalities in relation to access to sanitation services, two are bordering the core municipality of the RMBH, one in the north and one in the south.

The analysis of intersecting forms of inequality suggests that the consideration of isolated criteria is not sufficient to properly capture disparities between particularly favored and disadvantaged segments of the population. The results presented show that the accumulation of certain characteristics tends to exacerbate the susceptibility to deprivation of access to "at least basic" sanitation services, as the inequalities between the two profiles were much higher than those observed considering each criterion in isolation. Disregarding the overlapping of disadvantageous characteristics in the same individuals or groups is recurrent in studies focused on inequality and the methodological strategy presented in this article is a simple way to get around this problem. However, it is important to remember that the population profiles created do not cover the entire population. For this reason, this strategy should be used to complement disaggregation analyses by single criteria. It is worth mentioning that, as in the case of the disaggregation by urban–rural status, several municipalities have a very small number of people classified in the most favored profile and a much larger number in the disadvantaged profile, which means that care should be taken when making comparisons between these groups in some municipalities.

One limitation of this research is that, in adapting the JMP methodology to estimate levels of access to "at least basic" services, municipalities that use a lot of predominantly traditional latrines tend to be disadvantaged in comparison with other municipalities. Due to the lack of information about the presence of slabs or roofs in the Demographic Census, half of the traditional latrines were classified as unimproved facilities, as was done in the national JMP estimates [18]. Although in the RMBH as a whole only 8% of the population uses this type of system, in some municipalities, it is the most widespread in use.

This paper presents several innovations in comparison to a related work previously published [6]. First, it presents a conceptual innovation by adopting the categories and concepts proposed by JMP in 2017, the international reference with regard to the assessment and monitoring of water supply, sanitation, and hygiene. While in the referred publication the categories were based on the type of sanitation facilities, as established in the National Plan of Basic Sanitation (PLANSAB), the conceptual framework used in this article focuses on the levels of sanitation services, enabling international comparability. Second, this article presents methodological innovations by proposing a synthetic index that allows comparing the levels of inter-municipal inequality considering multiple criteria. Additionally, the consideration of intersecting forms of inequality made it possible to assess the impacts of overlapping disadvantageous characteristics with regard to access to sanitation services. Thirdly, this work makes progress in comparison to previous efforts to characterize access to sanitation facilities in the RMBH [6], showing the estimates of access calculated from the most recent household sample surveys produced by the IBGE. Finally, it explores the pace of expansion of access to "at least basic" services in the RMBH through a prospective exercise, projecting its universalization in each municipality based on the expansion rates observed in the 2000–2010 inter-census period.

## 5. Conclusions

One of the most important lessons drawn from the Millennium Development Goals (MDGs) was that progress should not be tracked only at the national level. Thus, the UN made a special effort to make monitoring at the local and regional level a priority in the case of the SDGs. They all have targets directly related to the responsibilities of local and regional governments, particularly their role in providing basic services, as in the case of sanitation services. By "localizing" target 6.2 of SDG 6, the present research has the potential to subsidize public policies in this field, contributing to the identification of population groups and priority areas of investment on a metropolitan and municipal scale. Focusing on these groups and areas can increase the effectiveness of inclusive social policies and optimize the use of public resources benefiting vulnerable populations in accordance with the SDG slogan: "leave no one behind". By trying to fill gaps highlighted by the World Health Organization (WHO) and the United Nations Children's Fund (UNICEF) indicated by the "Task Force on Monitoring Inequalities for the 2030 Sustainable Development Agenda" [24], the methodological procedures presented in this article are of potential interest in the efforts to "localize" the SDGs. These may involve the mobilization of different social actors and stakeholders, such as members of academia, civil society institutions, and agents involved in the provision and regulation of sanitation services.

**Author Contributions:** Conceptualization, R.C.d.C., M.I.P.N. and L.H.; methodology, R.C.d.C., M.I.P.N. and L.H.; software and visualization, R.C.d.C.; writing—original draft preparation, R.C.d.C.; writing—review and editing, M.I.P.N. and L.H.; supervision, L.H.; and funding acquisition, L.H. All authors have read and agreed to the published version of the manuscript.

**Funding:** This research was funded by Coordenação de Aperfeiçoamento de Pessoal de Nível Superior (CAPES) by grant funding to Rodrigo Coelho de Carvalho (PNPD Demografia 143/2019) and Maria Inês Pedrosa Nahas (PNPD Saúde Coletiva 086/2013). The APC was funded by Oswaldo Cruz Foundation (Fiocruz).

**Conflicts of Interest:** The authors declare no conflict of interest.

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
