# Peer review of "Localizing Sustainable Development Goal 6: An Assessment of Equitable Access to Sanitation in a Brazilian Metropolitan Region"

_sustainability, doi:10.3390/su12176776_

Round 1

Reviewer 1 Report

This study evaluated the inequalities in access to WASH in the Belo Horizonte 20 Metropolitan Region (RMBH) and proposed the effective evaluation and monitoring strategies for the target 6.2. Overall, this study treated an interesting topic, but the reviewer has several comments and clarifications requested below:

Abstract

  1. It seems that the authors use the 2030 Agenda and the SDGs interchangeably. Technically, the 2030 Agenda encompasses the 17 SDGs and their 169 targets (https://www.un.org/ga/search/view_doc.asp?symbol=A/RES/70/1&Lang=E). Thus, use the appropriate term (e.g. SDG 6) to enhance readability.
  2. Why did the authors adapt the JMP’s framework instead of the SDG indicators? (https://unstats.un.org/sdgs/indicators/Global%20Indicator%20Framework%20after%202020%20review_Eng.pdf)

Introduction

  1. The reason for having RMBH as a case study is not well addressed—please add general statistical figures on a systematic gap and highly deficient public service systems.
  2. Previous literature attempting to localize SDG 6 is missing.
  3. The benefits of improving WASH for public health should be included.
  4. The logic link of why the authors use municipalities as a level of investigation should be highlighted.

Methods

  1. Subsections explaining SDGs 6 and JMP can be included.
  2. A subsection dedicated to the study area can be beneficial for readers.
  3. A separate subsection for inclusion/exclusion criteria for the study population can be helpful.
  4. Data descriptions should exist in separate subsections.

Discussion

  1. The correlation between household income per capita and educational attainment should be clarified more by providing (at least general) statistics in Brazil.
  2. As indicated in target 6.2, Gender can be a strong factor for access to WASH; however, the authors did not consider gender in the analyses. Please clarify why.

Reviewer 2 Report

Overview and general recommendation:

This article seeks to contribute to the 17 definition of evaluation and monitoring strategies for the target 6.2 of the 2030 Agenda at the local 18 level, adapting the international criteria established by the Joint Monitoring Programme for Water 19 Supply, Sanitation and Hygiene (WHO/UNICEF) at the municipal scale, using the Belo Horizonte Metropolitan Region (RMBH) as a case study.

This problem is reflected in the absence of sanitation services and inequalities in access between population subgroups and among the 34 municipalities that currently comprise the region.

The article sought to explore the access to services by different population subgroups and by the municipalities that compose the RMBH. In addition to assessing socio-spatial inequalities, an estimate of the current situation in this region as a whole is presented based on the most recent data available and a projection of universal access to sanitation services according to international criteria, based on the trends observed in recent decades.

The paper is well structured and interesting for the review’s purposes.

The title accurately reflect the content and the abstract is complete and stand-alone.

Major comments:

Referring to the basic data used, it seems to me, fundamental, inserting a paragraph describing their source, their differences their availability, so that also people from different countries can understand opportunities and limits to use them.

In fact, as highlighted in the 346, 347, 348 lines of the paper: “The first results presented refer to access to sanitation facilities in the RMBH as a whole using the most recent data available from IBGE's household sample surveys”.

The, in the paper are used also the Joint Monitoring Programme for Water Supply, Sanitation and Hygiene 54 or JMP (WHO/UNICEF) data.

Besides, in the 401 and 402 lines of the paper it has been correctly specified that:

However, it is important to remember that the population profiles created do not cover the entire population. For this reason, this strategy should be used to complement disaggregation analyses by single criteria”.

Reviewer 3 Report

From the point of view of scientific innovation, the article is not very rich, but the topic is interesting and current and the study is replicable.

  1. In the graph in Figure 1, for example, the green line fluctuations are not logical. The authors state that it may be due to different methodological approaches, but they should try to improve or clarify this point - introducing corrective factors, moving averages or even considering more widely spaced values, corresponding to periods where equal methodologies are applied (every two years?);
  2. Bibliographic references are relatively shorten and the vast majority of citations refer to Brazilian articles. Can the authors enrich this aspect?

Round 2

Reviewer 1 Report

The manuscript was improved considerably. One thing the reviewer would like to recommend is to combine materials and methods as one section.

Author Response

We thank the reviewer for the feedback. We did a final spell check and we have amended the text as required, combining "Materials and Methods" in a single section.